# Expression of Heat Shock Protein 90 Genes Induced by High Temperature Mediated Sensitivity of *Aphis glycines* Matsumura (Hemiptera: Aphididae) to Insecticides

**DOI:** 10.3390/insects16080772

**Published:** 2025-07-28

**Authors:** Xue Han, Yulong Jia, Changchun Dai, Xiaoyun Wang, Jian Liu, Zhenqi Tian

**Affiliations:** Key Laboratory of Crop Pests in Northern Cold Regions of Heilongjiang Province, College of Plant Protection, Northeast Agricultural University, No. 600 Changjiang Road, Xiangfang District, Harbin 150030, China; hanxue3980@126.com (X.H.); jiayulong0706@163.com (Y.J.); changchundai@163.com (C.D.); wxy3236@126.com (X.W.); jliu@neau.edu.cn (J.L.)

**Keywords:** *Aphis glycines*, heat shock protein, thermal stress, insecticide resistance

## Abstract

Heat shock protein (HSP) genes are known to be activated in response to abiotic stresses. The *Hsp90* genes in *Aphis glycines* Matsumura (Hemiptera: Aphididae) (*AgHsp75*, *AgHsp83*, and *AgGrp94*) are significantly upregulated under both high-temperature conditions and insecticide exposure. The mortality of *A. glycines* was significantly increased after silencing *AgHsp90s*, indicating their critical role in stress tolerance. These findings suggest that targeting HSPs pathways may provide novel strategies for sustainable aphid management in the context of climate warming.

## 1. Introduction

With global climate change, the frequency of extreme weather events is increasing, and their duration is becoming prolonged [1]. Recent observations indicate a warming trend in the local climate of Harbin, northeastern China, with diurnal high temperatures frequently exceeding 35 °C for approximately seven consecutive days during summer and autumn in Heilongjiang Province [2]. This phenomenon significantly impacts poikilothermic animals, particularly insects [3]. High temperature exhibits negative effects on native insect herbivores, including reduced survival rates, compromised reproductive capacity, and diminished food intake, ultimately impairing growth and development [4,5]. Extreme high temperatures (EHTs) can induce protein denaturation and heat damage, manifesting as altered phospholipid membrane fluidity and disrupted cellular homeostasis [6]. Furthermore, EHTs promote the accumulation of reactive oxygen species (ROS) in insects, leading to oxidative stress (OS) and resulting in damage to DNA, lipids, or proteins [7].

Insects have evolved complex heat tolerance mechanisms to cope with thermal stresses, including behavioral, physiological, biochemical, and molecular adaptations [8,9]. Among these, heat shock proteins (HSPs) serve as essential molecular chaperones that participate in diverse biological processes. They facilitate the refolding of misfolded proteins, thereby maintaining cellular proteostasis [10]. HSPs have been implicated as critical determinants of both insect thermotolerance and insecticide resistance, orchestrating stress response pathways [11,12]. Based on molecular weight and sequence homology, HSPs are classified into six superfamilies: small HSPs (sHSPs), HSP40, HSP60, HSP70, HSP90, and HSP100 [13,14]. While HSP70 has been extensively studied in insect thermoregulation, the description of HSP90 in thermal adaptation remains comparatively underexplored [15]. In this study, we focused on three members of the *AgHsp90s*: *AgHsp75*, *AgHsp83*, and *AgGrp94*. *AgHsp83* represents the canonical cytosolic *Hsp90* homolog. Although designated *AgHsp75* (75 kDa), it shares high homology in core domains (notably the conserved ATPase domain) with *AgHsp83* and contains signature *Hsp90* motifs, confirming its classification within the *Hsp90* family despite nominal molecular weight variations across species. *AgGrp94* (Glucose-Regulated Protein 94) denotes the endoplasmic reticulum (ER)-resident *Hsp90* isoform (also termed *Hsp90b1*), primarily involved in folding, assembly, and quality control of ER luminal proteins, and its expression is often induced by ER stress (e.g., unfolded protein response) [16]. Therefore, verifying the functions of these three genes can provide a basis for in-depth research on aphid *Hsp90s*.

The soybean aphid, *Aphis glycines*, is a major pest of cultivated soybean. It causes leaf shrinkage, plant dwarfing, poor root development, reduced pod number, and decreased hundred-grain weight by feeding on the phloem sap of soybean plants. Severe infestations can lead to plant wilting and mortality [17,18]. *Aphis glycines* can also transmit plant viruses, such as soybean mosaic virus [19], alfalfa mosaic virus [20], and potato virus Y [21]. Additionally, aphids excrete honeydew on plant leaves, leading to sooty mold, which obstructs leaf photosynthesis [18].

The escalating frequency and dosage of chemical insecticide applications have led to the development of robust insecticide resistance in insect populations. A growing body of evidence indicates that high-temperature stress significantly enhances insecticide resistance in insects [22,23,24]. Notably, recent studies suggest that HSP70 in aphids may serve as critical mediators linking thermal stress responses to insecticide resistance mechanisms [12,25,26]. For instance, the high-temperature induced upregulation of *Hsp70* genes in *Aphis gossypii* and *A. glycines* has been correlated with increased resistance to multiple insecticides, including acetamiprid, sulfoxaflor, and imidacloprid. Specifically, in *A. glycines*, imidacloprid treatment significantly elevated the relative expression of *Hsp75* (a member of *Hsp90s*), suggesting its potential involvement in insecticide resistance [27]. Nevertheless, the role of *Hsp90* genes in regulating insecticide sensitivity of *A. glycines* remains poorly characterized.

In this study, based on the results of transcriptome sequencing, three *Hsp90* genes were selected. We examined the expression patterns of *Hsp90* genes to elucidate their role in mediating the relationship between thermal stress and insecticide resistance in *A. glycines* using reverse transcription quantitative PCR (RT-qPCR). Furthermore, the role of *Hsp90* genes (*AgHsp75*, *AgHsp83*, and *AgGrp94*) in sensitivity of *A. glycines* to imidacloprid and lambda-cyhalothrin was identified using RNA interference (RNAi) technology. Our findings provide novel insights into the physiological adaptation mechanisms of *A. glycines* under thermal stress and insecticide exposure and establish a theoretical framework for developing molecular biotechnology-based control strategies targeting this pest under elevated temperature conditions.

## 2. Materials and Methods

### 2.1. Aphid and Host Source

*Aphis glycines* were collected from a soybean field at Northeast Agricultural University (NEAU), Harbin, Heilongjiang Province, China (126.72° E, 45.74° N) in 2021. The soybean cultivar Heinong 51 (Fangyuan Agricultural Co., Ltd., Wuchang, Heilongjiang Province, China) was used for aphid rearing. A single apterous adult aphid was transferred to a soybean plant at the V2 growth stage. The aphids and plants were maintained in a nylon mesh insect-rearing cage (40 × 40 × 40 cm) at 25 ± 1 °C, 70 ± 5% RH, and a photoperiod of 16: 8 (L: D) h in an RTOP-D Intelligent Climate Chamber (Zhejiang TOP Cloud-agri Technology Co., Ltd., Hangzhou, China).

### 2.2. Total RNA Extraction, cDNA Synthesis, and Gene Cloning

The total RNA of aphids was extracted from aphids using Trizol reagent (Invitrogen, Carlsbad, CA, USA) according to the manufacturer’s instructions. RNA quality and concentration were assessed using a NanoDrop spectrophotometer (Thermo Fisher Scientific, Waltham, MA, USA). First-strand cDNA was synthesized using 1 μg total RNA and the Fast First-Strand cDNA Synthesis Mix for RT kit (Goonie Biotech Co., Ltd., Guangzhou, China). The primers were designed based on the predicted ORFs using primer premier 5.0 (Premier Biosoft International, Palo Alto, CA, USA; Appendix A). The amplification products were purified from 1% agarose gels using Thermo Scientific GeneJET Kit (Thermo Fisher Scientific, Waltham, MA, USA). The purified fragments were cloned into the pEASY-T3 Simple vector (TransGen Biotech, Beijing, China) and transformed into chemically competent Trans1-T1 Escherichia coli cells (TransGen Biotech, Beijing, China). Positive clones were selected using blue-white screening and subsequently sequenced by Sangon Biotech Co., Ltd. (Shanghai, China). Plasmids containing correctly sequenced inserts were used as templates for dsRNA synthesis.

### 2.3. Real-Time Quantitative PCR (RT-qPCR) Analysis

The quantitative real-time (qRT-) PCR was performed using the Hieff qPCR SYBR Green Master Mix kit (Yeasen Biotechnology Co., Ltd., Shanghai, China) on a Bio-Rad CFX Maestro system (BioRad Co., Ltd., Hercules, CA, USA). The thermal cycling conditions consisted of an initial denaturation at 95 °C for 5 min, followed by 40 cycles of 95 °C for 10 s, 60 °C for 30 s, and 72 °C for 20 s, with a final extension at 72 °C for 10 min. Melting curve analysis was performed within the 60–95 °C temperature range to confirm the consistency and specificity of each reaction product. The qRT-PCR primers of *AgHsp75*, *AgHsp83*, and *AgGrp94* were designed using Beacon Designer 7 and are listed in Appendix A. The *EF1α*, previously validated as stably expressed in *A. glycines*, was used as the reference gene [28]. Relative gene expression levels were calculated using the 2^−∆∆Ct^ method [29,30]. Each experiment included three biological replicates with three technical replicates per sample.

### 2.4. Gene Expression Responds to Thermal Treatments

In this study, the newly born nymphs were initially reared at 25 °C. The 1st, 2nd, 3rd, and 4th instar nymphs, along with 3-day-old adults, were transferred to 29 °C and 33 °C for 24 h thermal treatments, respectively. Each developmental stage was triplicated at each temperature treatment. Following thermal treatment, ten aphids from each treatment group were collected into 1.5 mL centrifuge tubes, immediately frozen in liquid nitrogen, and subsequently stored at −80 °C until further analysis.

### 2.5. Gene Expression Responds to Insecticide Treatment

In this study, imidacloprid (active ingredient 98% *w*/*w*, Binnong Technology Co., Ltd., Binzhou, Shandong Province, China) and lambda-cyhalothrin (active ingredient 98% *w*/*w*, Binnong Technology Co., Ltd., Binzhou, Shandong Province, China) were dissolved in dimethylformamide (DMF) to prepare 100 mg/mL stock solutions. Then the stock solutions were diluted to 5 concentration gradients using 0.1% Triton X-100. A 0.1% Triton X-100 solution and ddH_2_O were used as the controls. Each concentration was tested with three biological replicates. For the bioassay, soybean leaves were dipped in each solution and air dried. Twenty-one-day-old apterous adult *A. glycines* were transferred onto the treated leaves and maintained at 25 ± 1 °C. Aphid mortality was recorded after 24 h. The median lethal concentration (LC_50_) for imidacloprid and lambda-cyhalothrin was calculated from mortality data using SPSS 27 (IBM Corp., Armonk, NY, USA, 2020).

One-day-old apterous adults were exposed to LC_30_ of imidacloprid for 24 h using the leaf-dip method. Treated aphids were then collected and immediately stored at −80 °C for subsequent RT-qPCR analysis of *ApHsp90* gene expression.

### 2.6. RNA Interference

The specific fragments of *AgHsp75*, *AgHsp83*, and *AgGrp94* were amplified from cDNA templates and subsequently used for dsRNA synthesis. The gene-specific primers containing T7 RNA polymerase promoter sequences are listed in Appendix A. The dsRNA was synthesized using the MEGAscrip T7 Kit (Invitrogen, Carlsbad, CA, USA) following the manufacturer’s protocol. Green fluorescent protein (GFP) dsRNA was synthesized as a negative control.

The RNAi was carried out according to the artificial diet rearing method as described by Wang et al. [31] with several modifications. The dsRNA (150 ng/μL) was mixed with 0.5 M sucrose solution to prepare the RNAi diet. Each experimental group consisted of 30 one-day-old apterous adult aphids, which were transferred to 4.5 cm diameter plastic tubes (2 open ends). One tube end was sealed with two layers of parafilm, while the other end was covered with gauze for ventilation. A 100 μL aliquot of the RNAi diet was applied to the center of the parafilm layer. A parallel control group was established using an artificial diet containing ds*GFP*. After 48 h of feeding, aphids were collected for RT-qPCR analysis to evaluate knockdown efficiency.

### 2.7. Bioassays

To explore the potential function of *AgHsp75*, *AgHsp83*, and *AgGrp94* in insecticide sensitivity, three-day-old adults were selected for experiments. Because these three *AgHsp90s* are expressed differently in response to high temperatures at different developmental stages, but they are all significantly highly expressed at three-day-old adults. The *A. glycines* that had been fed on dsRNA were exposed to LC_50_ concentrations of imidacloprid and lambda-cyhalothrin using the leaf-dip method, as mentioned above. Mortality was assessed after 24 h of exposure. Each bioassay was independently repeated three times (biological replicates), with 20 individuals tested per replication.

### 2.8. Data Analysis

The Shapiro–Wilk test was employed to test the normality of the data. In the absence of normality, the data were log10(x + 1) transformed. The relative expression levels of *AgHsp90s* at different developmental stages of *A. glycines* were analyzed using one-way ANOVA followed by Tukey’s Honestly Significant Difference (HSD) test for post hoc multiple comparisons. Student’s *t*-test was used to analyze the differences between two samples, including RNA interference efficiency and the expression level of *AgHsp90s* after exposure to insecticides. The mortality rate after RNA interference was analyzed using the Mann–Whitney U test. The probit regression was used to simulate and fit the virulence regression equation. All statistical analyses were performed using SPSS 27 (IBM Corporation, Armonk, NY, USA).

## 3. Results

### 3.1. AgHsp90 Genes Profiling at Different Developmental Stages

The relative expression levels of *AgHsp75*, *AgHsp83*, and *AgGrp94* varied significantly across different developmental stages. Both *AgHsp75* (*F* = 33.31; df = 4; *p* < 0.05; Figure 1a) and *AgHsp83* (*F* = 8.24; df = 4; *p* < 0.05; Figure 1b) exhibited the highest during the 2nd and 4th instar nymphal stages, respectively, showing significant upregulation compared to the 1st instar nymphs. Notably, *AgGrp94* demonstrated its highest expression level specifically in the 2nd instar nymphs (*F* = 96.39; df = 4; *p* < 0.05; Figure 1c).

### 3.2. AgHsp90 Genes Profiling in Response to Different Thermal Stresses

The relative expression levels of *AgHsp75*, *AgHsp83*, and *AgGrp94* were significantly upregulated following heat stress (*p* < 0.05; Figure 2). Heat stress induced a generally dose-dependent upregulation pattern, with expression levels increasing proportionally to temperature elevation during each developmental stage. Notably, the most pronounced induction of *Hsp90* gene expression occurred in the 1st instar nymphs and adult aphids. Specifically, the expression of *AgHsp75* at 33 °C exhibited a remarkable upregulation, reaching 53.15-fold and 24.29-fold relative to the control group (*p* < 0.05; Figure 2a).

### 3.3. Expression Levels of AgHsp90 Genes Following Insecticide Exposure

The LC_30_ values for imidacloprid and lambda-cyhalothrin are presented in Table 1. Exposure to LC_30_ concentrations of imidacloprid and lambda-cyhalothrin significantly upregulated *ApHsp90* gene expression (*p* < 0.05; Figure 3). Specifically, following 24 h of imidacloprid exposure, the expression levels of *AgHsp75*, *AgHsp83*, and *AgGrp94* increased by 7.87-fold, 3.52-fold, and 3.65-fold, respectively. Similarly, lambda-cyhalothrin exposure induced significant upregulation of *AgHsp90s*, with mRNA levels increasing to 1.41-fold, 1.33-fold, and 3.04-fold for *AgHsp75*, *AgHsp83*, and *AgGrp94*, respectively, compared to the control group.

### 3.4. Effect of RNAi on AgHsp90 Gene Expression and Bioassays

In order to evaluate the interference efficiency, the relative expression levels of *AgHsp90* genes were examined at 24 h, 48 h, and 72 h post-feeding. The highest interference efficiency was observed at 48 h post-feeding, followed by a significant decline at 72 h (*p* < 0.05; Figure 4).

Based on the above results, we assessed insecticide sensitivity at the peak silencing time point (48 h). Mortality rates of *A. glycines* exposed to imidacloprid and lambda-cyhalothrin for 24 h after *AgHsp90s* silencing were significantly increased compared to control groups (*p* < 0.05; Figure 5). Following feeding ds*AgHsp75*, ds*AgHsp83*, and ds*AgGrp94*, the mortality of *A. glycines* treated with imidacloprid increased by 37%, 22%, and 32%, respectively (Figure 5a). Similarly, mortality of *A. glycines* exposed to lambda-cyhalothrin increased by 22%, 18%, and 18%, respectively (Figure 5b).

## 4. Discussion

Global climate change is characterized by a significant increase in both the frequency and intensity of extreme high-temperature (EHT) events, which affect insects across multiple biological levels. EHTs can cause direct thermal damage while simultaneously inducing a cascade of molecular, biochemical, and physiological responses. HSPs represent a primary cellular defense mechanism, functioning to synthesize and accumulate specific chaperone molecules that prevent protein denaturation and cellular dysfunction under EHT conditions [32]. Previous studies have demonstrated that the transcriptional profiles of insect *Hsp* genes are developmentally regulated. Consistent with these findings, our results revealed distinct developmental expression patterns for the three *Hsp90* genes in *A. glycines* (Figure 1). Specifically, *AgHsp75* and *AgHsp83* exhibited peak expression levels during the N2 and N4 nymphal stages (Figure 1a,b), while *AgGrp94* showed highest expression at the N2 stage (Figure 1c). The expression patterns of *Hsp70* genes displayed similar developmental trends to those observed for *Hsp90* genes. In *A. gossypii*, expression level of *Hsp70* gene was significantly higher in the 2nd-instar nymphs compared to other developmental stages [11]. In contrast, *Hsp70* expression peaked in adult *Myzus persicae* [33]. In *Rhopalosiphum padi*, all four analyzed *Hsp* genes exhibited distinct developmental expression profiles [34]. Furthermore, *Hsp70* demonstrated stage-specific expression variation in *A. gossypii*, with highest levels in 4th-instar nymphs relative to 1st- and 2nd-instar nymphs and adults [35].

The expression of *Hsp* genes exhibits stage-specific responses to high-temperature stress. When *A. glycines* were exposed to 29 °C and 33 °C, all three *Hsp90* genes (*AgHsp75*, *AgHsp83*, and *AgGrp94*) demonstrated significant upregulation (Figure 2). Especially, the temperature-induced expression of *AgHsp90* genes was most pronounced in 1st-instar nymphs and adult aphids. This differential expression pattern may reflect distinct physiological requirements at different developmental stages. For instance, the 1st-instar nymphs are more thermally vulnerable; they may require the activation of higher levels of heat shock proteins to protect body tissues from heat damage [12,36]. The elevated expression in adults may be crucial for maintaining reproductive function under thermal stress [37,38]. These findings suggest that *Hsp90* genes play stage-specific adaptive roles in thermotolerance, which may contribute to surviving and reproducing across different developmental stages under temperature stress.

Given the diverse biological functions of HSPs, they serve as critical molecular chaperones that mediate cellular responses to various exogenous stresses, including high temperatures, insecticides, and other abiotic factors. For instance, the differential expression patterns of *AcHsp83a* and *AcHsp83b* in *Arma chinensis* suggest that these genes are pivotal for growth, development, and stress tolerance, particularly under extreme temperatures and UV-B radiation [39]. Consistent with previous findings, our study demonstrates that *Hsp75* expression in *A. glycines* is inducible by both thermal stress and insecticide exposure [27], further highlighting the multifaceted role of HSPs in stress adaptation. Due to the expression level of *Hsp90s* being different during different stages, there may be variations in insecticide resistance. Our results reveal that *Hsp90* genes are significantly upregulated in three-day-old adults *A. glycines* upon exposure to insecticides, underscoring their potential involvement in stress response pathways. Similarly, there are also many reports on its role in insecticide resistance. This observation aligns with extensive evidence documenting the role of HSP90 family members in insecticide resistance. For instance, *Sogatella furcifera* exhibits upregulation of seven *Hsp* genes under imidacloprid stress [40]. While similar responses have been reported in *Leptinotarsa decemlineata* [41], *Cydia pomonella* [42], and *Liposcelis bostrychophila* [43]. Notably, in *Anopheles arabiensis*, heat shock not only exacerbates existing pyrethroid resistance but also induces de novo resistance phenotypes [44]. Functionally, our study provides direct evidence that knockdown of *AgHsp90* genes significantly increases *A. glycines* susceptibility to insecticides, thereby confirming their contributory role in insecticide resistance. This finding is corroborated by similar observations in *Nilaparvata lugens*, where *Hsp70* knockdown significantly enhances insecticide sensitivity [26]. Collectively, these results emphasize that diverse HSP family members play indispensable roles in insect adaptation to both abiotic stresses and xenobiotic challenges, particularly in the context of insecticide resistance mechanisms.

Previous studies have suggested that organisms can exhibit cross-protection against multiple stressors, wherein exposure to one stressor induces resistance to another. For instance, *Bemisia tabaci* exposed to thiamethoxam exhibited reduced mortality and an increased lethal mean time (LT_50_) [45]. Similarly, sublethal doses of neonicotinoid insecticides enhanced the thermal tolerance and survival rates of *Apis mellifera* [46]. In this study, we found that high temperatures upregulated *Hsp90* gene expression, which modulates the sensitivity of *A. glycines* to insecticides. These findings suggest that *Hsp90* genes play a regulatory role in cross-protection mechanisms. This phenomenon may extend to other insect species, as evidenced by studies in *Tetranychus cinnabarinus*, where *TcHsp90* was implicated in cross-protection against thermal stress and emamectin benzoate exposure [47]. Additionally, *Apolygus lucorum* exhibited *Hsp90*-mediated responses to combined temperature and insecticide stresses [48]. Collectively, these studies highlight the critical role of *Hsp90* genes in conferring cross-resistance between high temperatures and insecticides in insects.

The emergence of cross-resistance significantly complicates pest management strategies. In summer conditions characterized by high temperature and humidity, field populations of *A. gossypii* predominantly exhibit the summer morphotype, which is characterized by smaller body size. Research has demonstrated that the summer morphotype displays enhanced resistance to insecticides [49]. This resistance is further exacerbated by the extensive application of chemical insecticides, rendering the *A. gossypii* control increasingly challenging [50]. Liu et al. [12] proposed that the upregulated expression of *Hsp70* genes in *A. gossypii* plays a critical role in this phenomenon. *Hsp70* genes mainly resist environmental stress by recognizing and transiently binding exposed hydrophobic residues [51]. In contrast, *Hsp90* likely operates further upstream within the stress response network. Rather than directly interacting with insecticides, *Hsp90* functions as a critical regulator of signal transduction cascades and a stabilizer of client proteins essential for survival under stress [52]. This functional distinction suggests potential synergies, where *Hsp90*-mediated stabilization enables *Hsp70* function. Notably, *A. glycines* exhibits parallel physiological responses to high temperatures, including the emergence of smaller summer morphotypes [53]. Based on these observations, we hypothesize that *Hsp90* genes, in addition to *Hsp70* genes, are also involved in regulating cross-resistance between high temperatures and insecticides in *A. glycines*. Given these findings, the control of *A. glycines* is expected to become progressively more difficult under ongoing climate warming. Consequently, elucidating whether *AgHsp90* genes serve as key regulators in the development of cross-resistance has important implications for the development of effective pest management strategies.

The interaction between insecticide toxicity and elevated temperature exhibits dynamic and complex responses. In the short term, high temperature often shows an antagonistic effect: on one hand, warming accelerates insecticide degradation (such as the environmental half-life of avermectin is significantly shortened) [54]; on the other hand, insects enhance their cellular protection capabilities through stress pathways, such as the induction of HSPs, thereby improving their tolerance to insecticides. Wang et al. [55] demonstrated that the LC_50_ of avermectin at 35 °C was significantly higher than at 25 °C in *Liriomyza trifolii*, supporting this antagonistic mechanism. However, under prolonged exposure, a synergistic effect gradually prevails. The combined stress of sub-lethal concentrations of insecticides and high temperature induces cross-adaptation, characterized by the coordinated upregulation of detoxification-related genes (P450, GST) and heat stress-related genes (*Hsp70*, *Hsp90*) [56]. Notably, avermectin-resistant strains have evolved constitutive heat resistance, exhibiting enhanced thermotolerance and an elevated critical maximum temperature for HSPs activation, along with increased baseline expression levels of heat shock proteins [57]. In our study, both high temperature and insecticide exposure induced the upregulation of *AgHsp90s*, suggesting that *A. glycines* may activate a shared stress-response pathway under these conditions. This finding implies the existence of an interactive resistance mechanism between high temperature and insecticides in *A. glycines*.

## 5. Conclusions

In conclusion, our results confirm that the three *Hsp90* genes in *A. glycines* not only play a role in the insect’s high-temperature response but also play an important role in its sensitivity to insecticides. Our discovery provides a perspective on the defense mechanisms of aphids against high temperatures and other environmental pressures. These data also indicate that *AgHsp90* genes may be a potential target for pest management based on RNAi.

## Figures and Tables

**Figure 1 insects-16-00772-f001:**
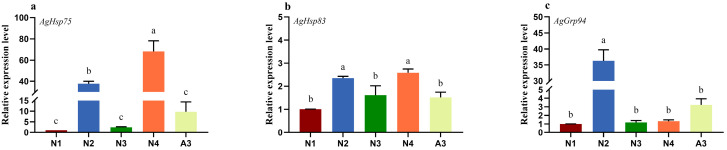
Developmental stage-specific expression patterns of three *AgHsp90* genes ((**a**) *AgHsp75*, (**b**) *AgHsp83*, and (**c**) *AgGrp94*) in *Aphis glycines*. Expression levels in instar nymphs (N1) were normalized to 1-fold. Data represent means ± SE (n = 3 biological replicates). Different lowercase letters indicate statistically significant differences among developmental stages (one-way ANOVA with Tukey’s HSD test, *p* < 0.05).

**Figure 2 insects-16-00772-f002:**
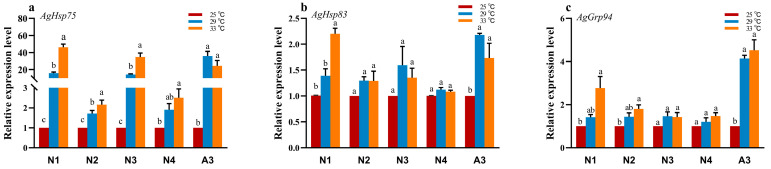
Temperature-dependent expression of *AgHsp90* genes in *Aphis glycines* after 24 h exposure to different high temperatures. (**a**) *AgHsp75*, (**b**) *AgHsp83*, and (**c**) *AgGrp94*. The 25 °C group served as the control. Different lowercase letters indicate significant differences among multiple samples (one-way ANOVA with Tukey’s HSD test, *p* < 0.05).

**Figure 3 insects-16-00772-f003:**
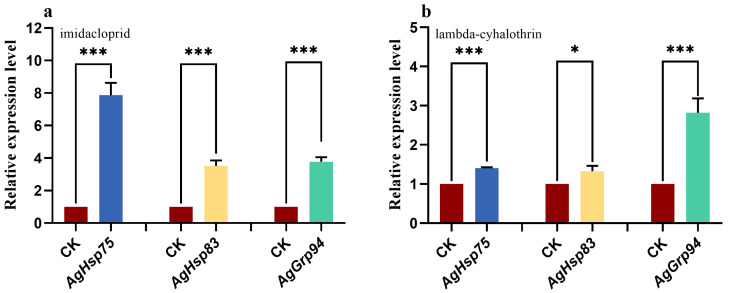
Insecticide-induced expression of *AgHsp90* genes in adult *Aphis glycines* after 24 h exposure. (**a**) Imidacloprid, (**b**) Lambda-cyhalothrin. Untreated insects served as the control. The asterisk above bar indicates significant differences (Student’s *t*-test, * *p* < 0.05, *** *p* < 0.001).

**Figure 4 insects-16-00772-f004:**
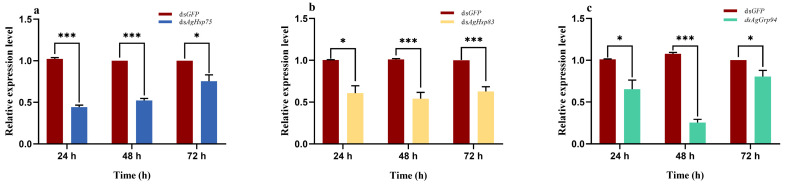
Expression of *AgHsp90* genes in adult *Aphis glycines* following dsRNA feeding. (**a**) *AgHsp75*, (**b**) *AgHsp83*, and (**c**) *AgGrp94*. The asterisk above bar indicates significant differences (Student’s *t*-test, * *p* < 0.05, *** *p* < 0.001).

**Figure 5 insects-16-00772-f005:**
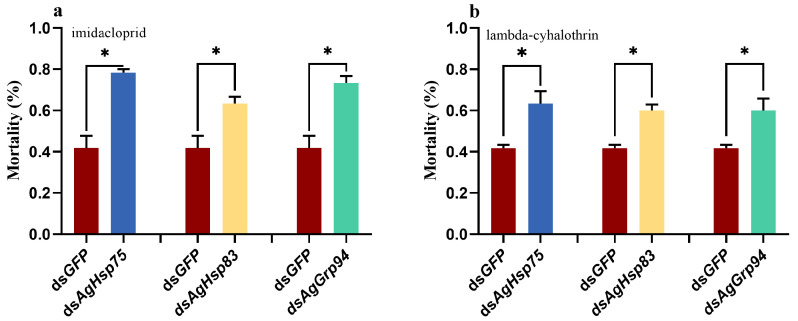
Mortality rates of *Aphis glycines* after 24 h exposure to insecticides following RNAi knockdown. (**a**) Imidacloprid, (**b**) Lambda-cyhalothrin. The asterisk above bar indicates significant differences (Mann–Whitney U test, * *p* < 0.05).

**Table 1 insects-16-00772-t001:** Resistance of *Aphis glycines* to different insecticides.

Insecticides	Slope (±SE)	LC_30_ (95% CL)(mg/L)	LC_50_ (95% CL)(mg/L)	χ^2^	df	*p*
Imidacloprid	1.59 ± 0.33	2.30 (0.90–4.02)	4.91 (2.60–8.54)	2.93	5	0.72
Lambda-cyhalothrin	1.51 ± 0.42	0.79 (0.11–1.62)	1.75 (0.55–3.18)	0.45	5	0.99

## Data Availability

The original contributions presented in this study are included in the article/Appendix A. Further inquiries can be directed to the corresponding author.

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
