# Peer review of "Expression of Heat Shock Protein 90 Genes Induced by High Temperature Mediated Sensitivity of Aphis glycines Matsumura (Hemiptera: Aphididae) to Insecticides"

_insects, 2025, doi:10.3390/insects16080772_

Round 1

Reviewer 1 Report

Comments and Suggestions for Authors

Totally, this research provides compelling mechanistic evidence that AgHsp90 genes underpin soybean aphid’s cross-resistance to heat and insecticides. It pioneers Hsp90 as a prime target for RNAi-mediated pest control and highlights urgent adaptations needed for chemical management under climate change. The results of this research is vital for the development of entomology, which should be accepted after minor revisions.

Here are my comments:

  1. Providing full scientific name of species at the first appearance in manuscript.
  2. Providing evidence about the correction of qPCR primers in Table S1, such as the amplification products of sequences.
  3. One-way ANOVA and Student’s t-test were used in this experiment, while these 2 tests are all parametric tests, which requiring the data all follow normal distribution. In fact, the mortality data always do not follow a normal distribution. Please provide evidence the data follow the normal distribution, or the authors should use nonparametric tests.
  4. In all figure legends, the species name appeared at the 1st time should be in full scientific name.
  5. Line 200, meaningless bold “temperature”.
  6. Line 195, wrong front of “℃”.
  7. Section 3.3, when you describe the results of Student’s t-test, you should provide the t-value and df of this analysis.
  8. Same comments for section 3.4.
  9. The Discussion dismisses Hsp70 (mentioned only once) despite its well-documented role in aphid insecticide cross-resistance. Failing to contrast Hsp90 vs. Hsp70 functions misses a key context. Authors should add a paragraph comparing *Hsp90/Hsp70* synergies/antagonisms. Example: "Unlike Hsp70 which directly binds neonicotinoids, Hsp90 may act upstream in stress signaling..
  10. The claim that Hsp90enables "cross-protection" (Ln 293-306) lacks experimental support. No combined stress tests (heat + insecticide) were performed to validate this.

Author Response

Comments 1: Providing full scientific name of species at the first appearance in manuscript.

Response 1:

The scientific name of species has revised in article.

Comments 2: Providing evidence about the correction of qPCR primers in Table S1, such as the amplification products of sequences.

Response 2:

The melt curve of AgHsp90s products has added in supplementary file. Please check figure S1. The single peak in the melting curve indicates that the qPCR primers produce highly specific products.

Comments 3: One-way ANOVA and Student’s t-test were used in this experiment, while these 2 tests are all parametric tests, which requiring the data all follow normal distribution. In fact, the mortality data always do not follow a normal distribution. Please provide evidence the data follow the normal distribution, or the authors should use nonparametric tests.

Response 3:

We have reanalyzed the mortality data using Mann-Whitney U tests and have incorporated this method into the Materials and Methods section. Please check Lines 192-193. And the figure 5 has revised after being reanalyzed.

Comments 4: In all figure legends, the species name appeared at the 1st time should be in full scientific name.

Response 4:

Thanks for your kindly comments. We had checked the species names in the article and changed the names of the species that 1st appeared to their full names.

Comments 5: Line 200, meaningless bold “temperature”.

Response 5:

We has revised bold temperature as regular temperature, please check Line 222.

Comments 6: Line 195, wrong front of “℃”.

Response 6:

We has revised “℃” as right front, please check Line 217.

Comments 7: Section 3.3, when you describe the results of Student’s t-test, you should provide the t-value and df of this analysis.

Response 7:

We had added these parameters in supplementary file. Please check Table S4.

Comments 8: Same comments for section 3.4.

Response 8:

We had added these parameters in supplementary file. Please check Table S5.

Comments 9: The Discussion dismisses Hsp70 (mentioned only once) despite its well-documented role in aphid insecticide cross-resistance. Failing to contrast Hsp90 vs. Hsp70 functions misses a key context. Authors should add a paragraph comparing *Hsp90/Hsp70* synergies/antagonisms. Example: "Unlike Hsp70 which directly binds neonicotinoids, Hsp90 may act upstream in stress signaling.

Response 9:

We had added this paragraph in Discussion. Please check in Lines 340-346.

Comments 10: The claim that Hsp90 enables "cross-protection" (Ln 293-306) lacks experimental support. No combined stress tests (heat + insecticide) were performed to validate this.

Response 10:

Thanks for your kindly comment. We have discussed this issue in Discussion section. Please check Lines 354-372.

Reviewer 2 Report

Comments and Suggestions for Authors

This manuscript presents valuable findings on the role of Aphis glycines Hsp90 genes in response to heat stress and insecticide exposure, highlighting potential targets for pest control under climate change scenarios. However, several aspects of the study require substantial improvement before it can be considered for publication. The Materials and Methods section lacks important experimental details that are necessary for reproducibility. The Results section does not present sufficient statistical detail or complete data to fully support the authors’ conclusions. Furthermore, the Discussion section should be expanded to contextualize the findings within a broader ecological and pest management framework, especially regarding the potential interaction between thermal stress and insecticide resistance.

1. The manuscript does not clearly justify the selection of the three AgHsp90 genes. The authors should elaborate on the rationale for choosing these specific genes in the Introduction section, ideally supported by relevant references or transcriptomic data.

2. The bioassay methods require clarification. The developmental stage of A. glycines used in the assay should be clearly stated. And the authors should clarify the choice of bioassay method in relation to the biology of the target stage and provide appropriate references.

3. In the Results section, the phrase ‘short-term heat stress’ is used to describe a 24-hour treatment. This duration may not be considered ‘short-term’ in insect physiology.

4. In all ANOVA analyses, F-values, degrees of freedom (between- and within-group) should be reported. For t-tests, the corresponding t-values and degrees of freedom should also be included. Please clarify whether assumptions such as normality and homoscedasticity were tested prior to these analyses. Please provide the results of normal distribution analysis in your research.

5. The methodology for LC30 and LC50 calculation is not described. The authors should specify whether probit or logit regression was used, and report the chi-square value and goodness-of-fit statistics (e.g., heterogeneity factor, significance level) for each estimate.

6. The study compares thermal tolerance between nymphs and adults but evaluates insecticide sensitivity only in adults. The authors should explain this discrepancy and consider discussing potential stage-specific differences in resistance phenotypes.

7. Figures 3 and 4 are the same. The RNAi knockdown efficiency over different time points is mentioned in the text but is not fully shown in the figures. Since the 48 h time point is highlighted as the most effective, quantitative knockdown data should be presented clearly.

8. Please confirm whether the 2−ΔΔCt method was used for relative expression quantification, and clearly indicate the reference gene employed. Additionally, clarify the normalizer used for comparative purposes (e.g., a reference line, baseline temperature, or untreated control), and indicate the unit or relative scale in figures and text.

9. Given that the study evaluates both heat stress and insecticide resistance, the potential interaction between these two factors should be discussed more thoroughly in the Discussion section. Are these effects synergistic or antagonistic? Consider expanding the discussion to include possible cross-resistance mechanisms, fitness trade-offs, or implications for pest management under climate change. Relevant literature that should be cited includes:

De Beeck LO, Verheyen J, Olsen K and Stoks R, 2017. Negative effects of pesticides under global warming can be counteracted by a higher degradation rate and thermal adaptation. J Appl Ecol, 54:1847–1855.

Wang Y C, Chang, Y W, Du, Y Z, 2021. Temperature affects the tolerance of Liriomyza trifolii to insecticide abamectin. Ecotox. Environ. Safe. 218:112307.

Wang Y C, Chang Y W, Du Y Z, 2021. Transcriptome analysis reveals gene expression differences in Liriomyza trifolii exposed to combined heat and abamectin exposure. PeerJ, 9: e12064.

Wang Y C, Chang Y W, Gong W R, et al., 2024. The development of abamectin resistance in Liriomyza trifolii and its contribution to thermotolerance. Pest Management Science, 2024, 80(4): 2053-2060.

Author Response

Comments 1: The manuscript does not clearly justify the selection of the three AgHsp90 genes. The authors should elaborate on the rationale for choosing these specific genes in the Introduction section, ideally supported by relevant references or transcriptomic data.

Response 1:

Thanks for your suggestion. The screening of these three genes was based on the previous transcriptome sequencing results. This section has been incorporated into the preface. Please check Lines 90-91.

Comments 2: The bioassay methods require clarification. The developmental stage of A. glycines used in the assay should be clearly stated. And the authors should clarify the choice of bioassay method in relation to the biology of the target stage and provide appropriate references.

Response 2:

The bioassay methods were previously described in detail, please see Lines 146-153. The developmental stage of A. glycines used in the assay has revised, please check Lines 178 -180. Since the bioassay methods have been described in detail before, no relevant literature was cited here.

Comments 3: In the Results section, the phrase ‘short-term heat stress’ is used to describe a 24-hour treatment. This duration may not be considered ‘short-term’ in insect physiology.

Response 3:

Thank you for your suggestion. We have already deleted “short-term”.

Comments 4: In all ANOVA analyses, F-values, degrees of freedom (between- and within-group) should be reported. For t-tests, the corresponding t-values and degrees of freedom should also be included. Please clarify whether assumptions such as normality and homoscedasticity were tested prior to these analyses. Please provide the results of normal distribution analysis in your research.

Response 4:

The F-values has added in main text and supplementary file. Please check Lines 198-203 and Table S2-5. And we added the normal distribution analysis method in Lines 186-187.

Comments 5: The methodology for LC30 and LC50 calculation is not described. The authors should specify whether probit or logit regression was used, and report the chi-square value and goodness-of-fit statistics (e.g., heterogeneity factor, significance level) for each estimate.

Response 5:

Thank you for pointing this out. We had added the chi-square value and goodness-of-fit statistics for each estimate. Please check Table 2. And we added the probit regression in Lines 193-194.

Comments 6: The study compares thermal tolerance between nymphs and adults but evaluates insecticide sensitivity only in adults. The authors should explain this discrepancy and consider discussing potential stage-specific differences in resistance phenotypes.

Response 6:

In this study, we mainly focus on the effect of high-temperature induced expression of AgHsp90s (AgHsp75, AgHsp83, and AgGrp94) on insecticide sensitivity of Aphis glycines. We found that these three AgHsp90s expressed differently to high temperatures at different developmental stages (Fig. 2). However, they all significantly highly expressed at three-day-old adults. Therefore, three-day-old adults were selected to evaluate insecticide sensitivity. The potential stage-specific differences in resistance phenotypes were also discussed in Discussion, please see Lines 300-302.

Comments 7: Figures 3 and 4 are the same. The RNAi knockdown efficiency over different time points is mentioned in the text but is not fully shown in the figures. Since the 48 h time point is highlighted as the most effective, quantitative knockdown data should be presented clearly.

Response 7:

Figure 4 has been changed to the correct version. Please check Line 244.

Comments 8: Please confirm whether the 2−ΔΔCt method was used for relative expression quantification, and clearly indicate the reference gene employed. Additionally, clarify the normalizer used for comparative purposes (e.g., a reference line, baseline temperature, or untreated control), and indicate the unit or relative scale in figures and text.

Response 8: In this study, we have mentioned that the relative gene expression levels were calculated using the 2-∆∆Ct method. Please see Lines 135-136. Additionally, we have also mentioned that the EF1α was selected as the reference gene, please see Lines 134-135. Because a previous study have validated its stable expression in A. glycines when subjected to various conditions, including temperatures and insecticides. Therefore, we cited this reference.

Comments 9: Given that the study evaluates both heat stress and insecticide resistance, the potential interaction between these two factors should be discussed more thoroughly in the Discussion section. Are these effects synergistic or antagonistic? Consider expanding the discussion to include possible cross-resistance mechanisms, fitness trade-offs, or implications for pest management under climate change. Relevant literature that should be cited includes:

De Beeck LO, Verheyen J, Olsen K and Stoks R, 2017. Negative effects of pesticides under global warming can be counteracted by a higher degradation rate and thermal adaptation. J Appl Ecol, 54:1847–1855.

Wang Y C, Chang, Y W, Du, Y Z, 2021. Temperature affects the tolerance of Liriomyza trifolii to insecticide abamectin. Ecotox. Environ. Safe. 218:112307.

Wang Y C, Chang Y W, Du Y Z, 2021. Transcriptome analysis reveals gene expression differences in Liriomyza trifolii exposed to combined heat and abamectin exposure. PeerJ, 9: e12064.

Wang Y C, Chang Y W, Gong W R, et al., 2024. The development of abamectin resistance in Liriomyza trifolii and its contribution to thermotolerance. Pest Management Science, 2024, 80(4): 2053-2060.

Response 9:

Thanks for your kindly comment. We have expanded the discussion on this part. Please check Lines 354-372.

Reviewer 3 Report

Comments and Suggestions for Authors

< !--StartFragment -->

The article is well written, and the research is accurate and interesting.
I made minor revisions for the following issues:

  • Figures 3 and 4 are identical; figure 4 is likely incorrect, as the text refers to time points at 24, 48, and 72 hours, which the figure does not match.
  • Line 268: duplicate period
  • Line 282: "while" is capitalized

< !--EndFragment -->

Author Response

Comments 1: Figures 3 and 4 are identical; figure 4 is likely incorrect, as the text refers to time points at 24, 48, and 72 hours, which the figure does not match.

Response 1:

Figure 4 has been changed to the correct version. Please check Line 244.

Comments 2: Line 268: duplicate period

Response 2:

Figure 1 and Figure 2 respectively illustrate the expression levels of Hsp90 in A. glycines at different stages under normal temperature conditions, as well as the expression levels of Hsp90 in A. glycines at different times under different high-temperature conditions. Figure 1 shows at which stage the expression level of the Hsp90 gene is the highest, which can provide a reference for determining the time period when using dsRNA to control A. glycines. Figure 2 more intuitively represents the responses of A. glycines at different ages under high-temperature conditions. We believe that this part of the data is necessary.

Comments 3: Line 282: "while" is capitalized

Response 3:

while has been revised as While, please check Line 307.
